

# Influence of Wind Turbine Design Parameters on Linearized Physics-Based Models in OpenFAST

Jason M. Jonkman[1], Emmanuel S. P. Branlard[1], John P. Jasa[1]

[1]National Renewable Energy Laboratory, 15013 Denver West Parkway, Golden, CO 80401, USA

*Correspondence to*: Jason M. Jonkman (jason.jonkman@nrel.gov)

**Abstract.** While most physics involved in wind energy are nonlinear, linearization of the underlying nonlinear wind-system equations is often important for understanding the system response and exploiting well-established methods and tools for analyzing linear systems. Linearized models are important for, e.g., eigenanalysis (to derive structural natural frequencies, damping ratios, and mode shapes) and controls design (based on linear state-space models). In controls co-design (CCD),
whose methods often rely on linearized time-domain models of the physics, the physical structure (often called the plant) and controller are designed and optimized concurrently, so, it is important to understand how changes to the physical design affect the linearized system. This work summarizes efforts done to understand the impact of design parameter variations in the physical system (mass, stiffness, geometry, etc.) on the linearized system using OpenFAST.

## 1 Introduction

The Aerodynamic Turbines Lighter and Afloat with Nautical Technologies and Integrated Servo-control (ATLANTIS) program funded by the U.S. Department of Energy (DOE) Advanced Research Projects Agency-Energy (ARPA-E) seeks to develop new technology pathways for the design of economically competitive floating offshore wind turbines (FOWTs) based on controls co-design (CCD) principles. Within ATLANTIS Topic Area 2 (Computer Tools), the National Renewable Energy Laboratory (NREL) leads a project developing the Wind Energy with Integrated Servo-control (WEIS) toolset, which is new
independent software that combines Wind-Plant Integrated System Design & Engineering Model (WISDEM®), OpenFAST (formerly known as FAST), and CCD functionality together with the goal of providing the offshore wind industry and research communities with an open-source, user-friendly, flexible tool to enable true CCD of the FOWT physical design together with the controller (Jonkman et al., 2021).

CCD methods, including those being implemented within WEIS, often rely on linearized time-domain models of the physics,
e.g., an optimal open-loop controller is solved with direct transcription based on a linearized time-domain model using quadratic programming, from which an optimal closed-loop controller can be derived for use in a higher fidelity nonlinear time-domain analysis. Here, the open-loop controller is optimized concurrently with the physical design (plant) via either a nested or simultaneous approach (Herber and Allison, 2018), so, it is important to understand how changes to the physical design affect the linearized system.



The open-source, physics-based engineering tool OpenFAST—developed by the NREL via support from DOE and applicable to the loads analysis of land-based on offshore fixed-bottom and floating wind turbines—has had the ability to generate linearized representations of the underlying nonlinear system, where the linearized system (state-space) matrices are valid only for small perturbations about an operating point and for a fixed set of design parameters (Jonkman, 2013; Jonkman and Jonkman, 2016; Jonkman et al., 2018; Jonkman et al., 2020). The linearized system is expressed in terms of Jacobians of the

state and output equations with respect to states and inputs. While OpenFAST can be linearized each time the structure is changed within the design iteration loop directly in a brute-force way, the operating point calculation and linearization are computationally intensive operations, so, this direct evaluation method is not necessarily the best method.

We originally considered computing Hessians—i.e., partial derivatives of the Jacobians with respect to design parameters— directly within OpenFAST, such that the linearized system (including changes to the operating point) could be written as a

function of design parameter perturbations. While some theoretical expressions were developed—both at the module and full-system levels of OpenFAST, including algebraic constraints—and while this approach would likely be computationally efficient with the design cycle, this effort was abandoned because it would have required major changes to OpenFAST. Regardless, the theoretical work does provide some physical insight and will be summarized in Sect. 2. The theory is also applied to a simple forced mass-spring-damper system as an illustrative example in Sect. 3.

Between the calculation of Hessians within OpenFAST and the direct evaluation (brute force) method of linearizing distinctly within every design iteration, is an intermediate method in terms of computational expense, whereby the linearized system is pre-computed for a range of design parameter variations, and these linearized matrices are interpolated within the design iteration loop to find a representative linear system for specific design parameter values. Both the direct evaluation method and interpolation method have been implemented in WEIS, which calls OpenFAST. The approaches used are explained in

Sect. 4 and a comparison between the results for a case study involving design parameter variations (tower density and stiffness, unstretched mooring line length) in the support structure of the International Energy Agency (IEA) Wind 15-MW reference wind turbine (Gaertner et al., 2020) atop the University of Maine semisubmersible (Allen et al., 2020) are presented in Sect. 5 to assess the quality of the intermediate method. The results are compared in terms of the impact of design parameter changes on the eigensolution (natural frequencies and damping) of the linearized continuous state matrix and in their computational

expense.

## 2 Theoretical Development

### 2.1 Nonlinear system and linearization without parameters

In the OpenFAST modularization framework, the most generalized nonlinear time-domain system implemented within a module that is still linearizable is given by Eq. (1) of Jonkman, 2013. For the sake of brevity and clarity in this paper, we

neglect possible discrete-time states and directly identify the set of parameters that characterize the system. The result is the semi-explicit differential algebraic equation (DAE) of index 1 represented mathematically as:



$$\dot{x} = X\left(x, z, u, t, p\right)$$
$$0 = Z\left(x, z, u, t, p\right) \text{ with } \left\|\frac{\partial Z}{\partial z}\right\| \neq 0 . \tag{1}$$
$$y = Y\left(x, z, u, t, p\right)$$

In Eq. (1), $x$ are the continuous states with first time derivatives $\dot{x}$ determined explicitly by the continuous-state functions $X(\ )$, $z$ are the constraint (algebraic) states determined implicitly by the constraint-state (algebraic) functions $Z(\ )$, $y$ are the module-level outputs determined explicitly by the output functions $Y(\ )$, $u$ are the module-level inputs (derived from the output of other modules), $p$ are the parameters that characterize the functions, and all terms are shown evaluated at time $t$. Examples of continuous states, constraint states, inputs, outputs, and parameters in wind turbine dynamics are given in Table 2 of Jonkman, 2013.

A linear representation of the nonlinear system from Eq. (1) is valid only for small deviations (perturbations) from an operating point (represented by $\left.\right|_{op}$). As shown in Jonkman, 2013, when holding the parameters fixed, each variable can be perturbed (represented by $\Delta$) about their respective operating point values,

$$x = x\big|_{op} + \Delta x$$
$$\dot{x} = \dot{x}\big|_{op} + \Delta \dot{x}$$
$$z = z\big|_{op} + \Delta z \ , \tag{2}$$
$$u = u\big|_{op} + \Delta u$$
$$y = y\big|_{op} + \Delta y$$

resulting in a linear time-invariant (LTI) system characterized by the state matrix $A$, the input matrix $B$, the state matrix for outputs $C$, and the input-transmission matrix for outputs $D$, all of which can be expressed in terms of Jacobians of the functions from Eq. (1) as follows:

$$\Delta \dot{x} = \underbrace{\left[\frac{\partial X}{\partial x} - \frac{\partial X}{\partial z}\left[\frac{\partial Z}{\partial z}\right]^{-1}\frac{\partial Z}{\partial x}\right]\Bigg|_{op}}_{A}\Delta x + \underbrace{\left[\frac{\partial X}{\partial u} - \frac{\partial X}{\partial z}\left[\frac{\partial Z}{\partial z}\right]^{-1}\frac{\partial Z}{\partial u}\right]\Bigg|_{op}}_{B}\Delta u$$

$$\Delta y = \underbrace{\left[\frac{\partial Y}{\partial x} - \frac{\partial Y}{\partial z}\left[\frac{\partial Z}{\partial z}\right]^{-1}\frac{\partial Z}{\partial x}\right]\Bigg|_{op}}_{C}\Delta x + \underbrace{\left[\frac{\partial Y}{\partial u} - \frac{\partial Y}{\partial z}\left[\frac{\partial Z}{\partial z}\right]^{-1}\frac{\partial Z}{\partial u}\right]\Bigg|_{op}}_{D}\Delta u \tag{3}$$



The constraint-state (algebraic) equations have been eliminated from the linearized system of Eq. (3) because, once linearized, the constraint-state equations can be easily solved for the perturbations of constraint states, $\Delta z$, shown in Eq. (4). Note that the requirement that the determinant of the Jacobian of the constraint-state function with respect to the constraint states, $\left|\dfrac{\partial Z}{\partial z}\right|$

, not be equal to zero from Eq. (1) means that the matrix inverse of the Jacobian from Eq. (3), $\left[\dfrac{\partial Z}{\partial z}\right]^{-1}$, exists and is bounded in the neighbourhood around a solution.

$$\Delta z = -\left[\left.\frac{\partial Z}{\partial z}\right|_{op}\right]^{-1} \left\{\left.\frac{\partial Z}{\partial x}\right|_{op} \Delta x + \left.\frac{\partial Z}{\partial y}\right|_{op} \Delta u\right\}. \tag{4}$$

Note also that the matrices in the linear state-space model are functions of the parameters $p$, but are fixed constants.

## 2.2 Linearization with respect to parameters, without constraints

Now we wish to understand the impact of small deviations in the parameters—representing the evolution of the design variables in the physical design (plant) optimization—on the linear state-space model as follows:

$$p = \left.p\right|_{op} + \Delta p. \tag{5}$$

We first present the approach without considering constraint (algebraic) states, so, $z$ and $Z(\ )$ are empty and are neglected. The formulation with constraint states is given in Sect. 2.3.

For clarity, only continuous parameters are considered, discrete parameters are neglected, and tensor notation is avoided, a specific element of a vector or matrix is given by a subscript after the variable, and the number of elements of each vector and matrix are written below each variable, with $N_x$ the number of continuous states, their first derivatives, and continuous-state functions; $N_z$ the number of constraint (algebraic) states and constraint-state (algebraic) functions ( $N_z = 0$ in this section and is nonzero in Sect. 2.3); $N_y$ number of module-level outputs and output functions; $N_u$ the number of module-level

inputs; and $N_p$ the number of parameters that have perturbations (this may be a subset of the total number of parameters). The linearization of the nonlinear time-domain system from Eq. (1) with respect to design parameters can be expressed in terms of Hessians. An example Hessian of the continuous-state functions with respect to parameters and continuous states evaluated at an operating point is written out in Eq. (6), where $n_x$ is a counter through each continuous state.



$$
\left. \frac{\partial^2 X_{n_x}}{\partial p \partial x} \right|_{op} _{N_p \times N_x} = \left[ \begin{array}{cccc} \dfrac{\partial^2 X_{n_x}}{\partial p_1 \partial x_1} & \dfrac{\partial^2 X_{n_x}}{\partial p_1 \partial x_2} & \cdots & \dfrac{\partial^2 X_{n_x}}{\partial p_1 \partial x_{N_x}} \\[2mm] \dfrac{\partial^2 X_{n_x}}{\partial p_2 \partial x_1} & \dfrac{\partial^2 X_{n_x}}{\partial p_2 \partial x_2} & & \dfrac{\partial^2 X_{n_x}}{\partial p_2 \partial x_{N_x}} \\[2mm] \vdots & & \ddots & \\[2mm] \dfrac{\partial^2 X_{n_x}}{\partial p_{N_p} \partial x_1} & \dfrac{\partial^2 X_{n_x}}{\partial p_{N_p} \partial x_2} & & \dfrac{\partial^2 X_{n_x}}{\partial p_{N_p} \partial x_{N_x}} \end{array} \right]_{op} _{N_p \times N_x} \quad \text{for } n_x = \{1, 2, \ldots, N_x\} \tag{6}
$$

Note that under the hypothesis of continuity of the second derivates, the order of differentiation does not matter, and the Hessian matrices are symmetric as illustrated in Eq. (7), where superscript $^T$ represents the matrix transpose.

$$
\left. \frac{\partial^2 X_{n_x}}{\partial p \partial x} \right|_{op} _{N_p \times N_x} = \left[ \left. \frac{\partial^2 X_{n_x}}{\partial x \partial p} \right|_{op} _{N_x \times N_p} \right]^T _{N_p \times N_x} \quad \text{for } n_x = \{1, 2, \ldots, N_x\} \tag{7}
$$

Including all parameter-related Hessians, as well as all nonlinear combinations of the parameter variations, $\Delta p$, the linearization of the nonlinear time-domain system from Eq. (1) with respect to design parameters, while neglecting constraints, 

is given by Eq. (8). The last term could be simplified if only linear contributions of $\Delta p$ are considered.

$$
\underset{N_x \times 1}{\Delta \dot{x}} = \underbrace{\left[ \left. \frac{\partial X}{\partial x} \right|_{op} _{N_x \times N_x} + \left[ \begin{array}{c} \underset{1 \times N_p}{\Delta p^T} \left. \dfrac{\partial^2 X_1}{\partial p \partial x} \right|_{op} _{N_p \times N_x} \\[2mm] \underset{1 \times N_p}{\Delta p^T} \left. \dfrac{\partial^2 X_2}{\partial p \partial x} \right|_{op} _{N_p \times N_x} \\[2mm] \vdots \\[2mm] \underset{1 \times N_p}{\Delta p^T} \left. \dfrac{\partial^2 X_{N_x}}{\partial p \partial x} \right|_{op} _{N_p \times N_x} \end{array} \right]_{N_x \times N_x} \right]}_{\substack{A(\Delta p) \\ N_x \times N_x}} \underset{N_x \times 1}{\Delta x} + \underbrace{\left[ \left. \frac{\partial X}{\partial u} \right|_{op} _{N_x \times N_u} + \left[ \begin{array}{c} \underset{1 \times N_p}{\Delta p^T} \left. \dfrac{\partial^2 X_1}{\partial p \partial u} \right|_{op} _{N_p \times N_u} \\[2mm] \underset{1 \times N_p}{\Delta p^T} \left. \dfrac{\partial^2 X_2}{\partial p \partial u} \right|_{op} _{N_p \times N_u} \\[2mm] \vdots \\[2mm] \underset{1 \times N_p}{\Delta p^T} \left. \dfrac{\partial^2 X_{N_x}}{\partial p \partial u} \right|_{op} _{N_p \times N_u} \end{array} \right]_{N_x \times N_u} \right]}_{\substack{B(\Delta p) \\ N_x \times N_u}} \underset{N_u \times 1}{\Delta u} + \underbrace{\left[ \left. \frac{\partial X}{\partial p} \right|_{op} _{N_x \times N_p} + \frac{1}{2} \left[ \begin{array}{c} \underset{1 \times N_p}{\Delta p^T} \left. \dfrac{\partial^2 X_1}{\partial p^2} \right|_{op} _{N_p \times N_p} \\[2mm] \underset{1 \times N_p}{\Delta p^T} \left. \dfrac{\partial^2 X_2}{\partial p^2} \right|_{op} _{N_p \times N_p} \\[2mm] \vdots \\[2mm] \underset{1 \times N_p}{\Delta p^T} \left. \dfrac{\partial^2 X_{N_x}}{\partial p^2} \right|_{op} _{N_p \times N_p} \end{array} \right]_{N_x \times N_p} \right]}_{\substack{X_p(\Delta p) \\ N_x \times N_p}} \underset{N_p \times 1}{\Delta p} \tag{8a}
$$



$$\Delta y_{N_y \times 1} = \underbrace{\left[ \left. \frac{\partial Y}{\partial x} \right|_{op} + \begin{bmatrix} \Delta p^T_{1 \times N_p} \left. \frac{\partial^2 Y_1}{\partial p \partial x} \right|_{op} \\ {}_{N_p \times N_x} \\ \Delta p^T_{1 \times N_p} \left. \frac{\partial^2 Y_2}{\partial p \partial x} \right|_{op} \\ {}_{N_p \times N_x} \\ \vdots \\ \Delta p^T_{1 \times N_p} \left. \frac{\partial^2 Y_{N_y}}{\partial p \partial x} \right|_{op} \\ {}_{N_p \times N_x} \end{bmatrix}_{N_y \times N_x} \right]}_{\substack{C(\Delta p) \\ N_y \times N_x}} \Delta x_{N_x \times 1} + \underbrace{\left[ \left. \frac{\partial Y}{\partial u} \right|_{op} + \begin{bmatrix} \Delta p^T_{1 \times N_p} \left. \frac{\partial^2 Y_1}{\partial p \partial u} \right|_{op} \\ {}_{N_p \times N_u} \\ \Delta p^T_{1 \times N_p} \left. \frac{\partial^2 Y_2}{\partial p \partial u} \right|_{op} \\ {}_{N_p \times N_u} \\ \vdots \\ \Delta p^T_{1 \times N_p} \left. \frac{\partial^2 Y_{N_y}}{\partial p \partial u} \right|_{op} \\ {}_{N_p \times N_u} \end{bmatrix}_{N_y \times N_u} \right]}_{\substack{D(\Delta p) \\ N_y \times N_u}} \Delta u_{N_u \times 1} + \underbrace{\left[ \left. \frac{\partial Y}{\partial p} \right|_{op} + \frac{1}{2} \begin{bmatrix} \Delta p^T_{1 \times N_p} \left. \frac{\partial^2 Y_1}{\partial p^2} \right|_{op} \\ {}_{N_p \times N_p} \\ \Delta p^T_{1 \times N_p} \left. \frac{\partial^2 Y_2}{\partial p^2} \right|_{op} \\ {}_{N_p \times N_p} \\ \vdots \\ \Delta p^T_{1 \times N_p} \left. \frac{\partial^2 Y_{N_y}}{\partial p^2} \right|_{op} \\ {}_{N_p \times N_p} \end{bmatrix}_{N_y \times N_p} \right]}_{\substack{Y_p(\Delta p) \\ N_y \times N_p}} \Delta p_{N_p \times 1} \quad (8b)$$

The parameter-dependent variations of the linear state matrices from Eq. (8) include the perturbed parameter form of the state matrix $A(\Delta p)$, the perturbed parameter form of the input matrix $B(\Delta p)$, the perturbed parameter form of the state matrix for outputs $C(\Delta p)$, and the perturbed parameter form of the input-transmission matrix for outputs $D(\Delta p)$. The additional terms at the end of each linear state-space equation from Eq. (8), $X_p(\Delta p)$ and $Y_p(\Delta p)$, cause offsets of the state and output perturbations as a result of the parameter variation (effectively representing the change in operating point as a result of the change in parameter). To derive this offset, let

$$
\begin{aligned}
\Delta x &= \left. \Delta x \right|_{op} + \Delta x' \\
\Delta \dot{x} &= \left. \cancel{\Delta \ddot{x}}\right|_{op}^{0} + \Delta \dot{x}' \\
\Delta u &= \left. \cancel{\Delta u}\right|_{op}^{0} + \Delta u' \\
\Delta y &= \left. \Delta y \right|_{op} + \Delta y'
\end{aligned}
\qquad (9)
$$

where $\left. \Delta x \right|_{op}$ represents the change in the continuous-state operating point associated with the parameter variations, $\left. \Delta y \right|_{op}$ represents the change in the module-level output operating point associated with the parameter variations, and the primed ($'$) variables represent the perturbations about the updated operating point. The change in continuous-state operating point associated with parameter variations is independent of time, so, $\left. \Delta \dot{x} \right|_{op} = 0$. Likewise, the module-level inputs are unaffected



by parameter variations, so, $\Delta u\big|_{op} = 0$. Equation (8a) can then be used to derive $\Delta x\big|_{op}$ from $\Delta p$. This $\Delta x\big|_{op}$ can then be

used to derive $\Delta y\big|_{op}$ using Eq. (8b), resulting in

$$\Delta x\big|_{op} = -A(\Delta p)^{-1} X_p(\Delta p)\Delta p$$
$$\Delta y\big|_{op} = \left[Y_p(\Delta p) - C(\Delta p) A(\Delta p)^{-1} X_p(\Delta p)\right]\Delta p \tag{10}$$

The final expressions for the linearization of the nonlinear time-domain system from Eq. (1) with respect to design parameters, while neglecting constraint states (represented by $\varnothing$), is given by Eq. (11), which is the parameterized form of Eq. (2), and Eq. (12), which is the parameterized form of Eq. (3).

$$x = x\big|_{op} + \Delta x\big|_{op} + \Delta x'$$
$$\dot{x} = \dot{x}\big|_{op} + \cancel{\Delta \dot{x}\big|_{op}}^{0} + \Delta \dot{x}'$$
$$z = \cancel{z\big|_{op} + \Delta z}^{\varnothing}$$


$$u = u\big|_{op} + \cancel{\Delta u\big|_{op}}^{0} + \Delta u' \tag{11}$$
$$y = y\big|_{op} + \Delta y\big|_{op} + \Delta y'$$
$$p = p\big|_{op} + \Delta p$$

$$\Delta \dot{x}' = A(\Delta p)\Delta x' + B(\Delta p)\Delta u'$$
$$\Delta y' = C(\Delta p)\Delta x' + D(\Delta p)\Delta u' \tag{12}$$

## 2.3 Linearization with respect to parameters, with constraints

The same process used in Sect. 2.2 can be applied when the underlying nonlinear system has constraint states—the equations just become more onerous, as shown in Eq. (13). As in Eq. (3), the constraint-state (algebraic) equations have been eliminated

from the linearized system of Eq. (13) because, once linearized, the constraint-state equations can be easily solved for the perturbations of constraint states, shown in Eq. (14), which is the parameterized form of Eq. (4). Hereby we assume





$$\left\lVert \left.\frac{\partial Z}{\partial z}\right\rvert_{op} + \begin{bmatrix} \underset{1\times N_p}{\Delta p^T} \left.\frac{\partial^2 Z_1}{\partial p \partial z}\right\rvert_{op} \\ \underset{N_p \times N_z}{} \\ \underset{1\times N_p}{\Delta p^T} \left.\frac{\partial^2 Z_2}{\partial p \partial z}\right\rvert_{op} \\ \underset{N_p \times N_z}{} \\ \vdots \\ \underset{1\times N_p}{\Delta p^T} \left.\frac{\partial^2 Z_{N_z}}{\partial p \partial z}\right\rvert_{op} \\ \underset{N_p \times N_z}{} \end{bmatrix} \right\rVert \neq 0$$

. The final expressions for the linearization of the nonlinear time-domain system from Eq. (1)

with respect to design parameters, while including constraint states, is still given by Eq. (11), which is the parameterized form of Eq. (2), and Eq. (12), which is the parameterized form of Eq. (3), except that $A(\Delta p)$, $B(\Delta p)$, $C(\Delta p)$, $D(\Delta p)$,

$X_p(\Delta p)$, and $Y_p(\Delta p)$ are given in Eq. (13) instead of Eq. (8) and the constraint states are no longer eliminated from Eq.

(11), i.e., $z = \left.z\right\rvert_{op} + \Delta z^{\varnothing}$ must be replaced with $z = \left.z\right\rvert_{op} + \Delta z$, with $\Delta z$ given by Eq. (14).





$$
\underset{N_x \times 1}{\Delta \dot{x}} = \left[ \underset{N_x \times N_x}{\left.\frac{\partial X}{\partial x}\right|_{op}} + \begin{bmatrix} \underset{1 \times N_p}{\Delta p^T} \left.\frac{\partial^2 X_1}{\partial p \partial x}\right|_{op} \\ \underset{N_p \times N_x}{} \\ \underset{1 \times N_p}{\Delta p^T} \left.\frac{\partial^2 X_2}{\partial p \partial x}\right|_{op} \\ \underset{N_p \times N_x}{} \\ \vdots \\ \underset{1 \times N_p}{\Delta p^T} \left.\frac{\partial^2 X_{N_x}}{\partial p \partial x}\right|_{op} \\ \underset{N_p \times N_x}{} \end{bmatrix}_{N_x \times N_x} - \left( \underset{N_x \times N_z}{\left.\frac{\partial X}{\partial z}\right|_{op}} + \begin{bmatrix} \underset{1 \times N_p}{\Delta p^T} \left.\frac{\partial^2 X_1}{\partial p \partial z}\right|_{op} \\ \vdots \\ \underset{1 \times N_p}{\Delta p^T} \left.\frac{\partial^2 X_{N_x}}{\partial p \partial z}\right|_{op} \end{bmatrix}_{N_x \times N_z} \right) \left( \underset{N_z \times N_z}{\left.\frac{\partial Z}{\partial z}\right|_{op}} + \begin{bmatrix} \underset{1 \times N_p}{\Delta p^T} \left.\frac{\partial^2 Z_1}{\partial p \partial z}\right|_{op} \\ \vdots \\ \underset{1 \times N_p}{\Delta p^T} \left.\frac{\partial^2 Z_{N_z}}{\partial p \partial z}\right|_{op} \end{bmatrix}_{N_z \times N_z} \right)^{-1} \left( \underset{N_z \times N_x}{\left.\frac{\partial Z}{\partial x}\right|_{op}} + \begin{bmatrix} \underset{1 \times N_p}{\Delta p^T} \left.\frac{\partial^2 Z_1}{\partial p \partial x}\right|_{op} \\ \vdots \\ \underset{1 \times N_p}{\Delta p^T} \left.\frac{\partial^2 Z_{N_z}}{\partial p \partial x}\right|_{op} \end{bmatrix}_{N_z \times N_x} \right) \right] \underset{N_x \times 1}{\Delta x}
$$

$$
\underset{A(\Delta p) \atop N_x \times N_x}{}
$$

$$
+ \left[ \underset{N_x \times N_u}{\left.\frac{\partial X}{\partial u}\right|_{op}} + \begin{bmatrix} \underset{1 \times N_p}{\Delta p^T} \left.\frac{\partial^2 X_1}{\partial p \partial u}\right|_{op} \\ \vdots \\ \underset{1 \times N_p}{\Delta p^T} \left.\frac{\partial^2 X_{N_x}}{\partial p \partial u}\right|_{op} \end{bmatrix}_{N_x \times N_u} - \left( \underset{N_x \times N_z}{\left.\frac{\partial X}{\partial z}\right|_{op}} + \begin{bmatrix} \underset{1 \times N_p}{\Delta p^T} \left.\frac{\partial^2 X_1}{\partial p \partial z}\right|_{op} \\ \vdots \\ \underset{1 \times N_p}{\Delta p^T} \left.\frac{\partial^2 X_{N_x}}{\partial p \partial z}\right|_{op} \end{bmatrix}_{N_x \times N_z} \right) \left( \underset{N_z \times N_z}{\left.\frac{\partial Z}{\partial z}\right|_{op}} + \begin{bmatrix} \underset{1 \times N_p}{\Delta p^T} \left.\frac{\partial^2 Z_1}{\partial p \partial z}\right|_{op} \\ \vdots \\ \underset{1 \times N_p}{\Delta p^T} \left.\frac{\partial^2 Z_{N_z}}{\partial p \partial z}\right|_{op} \end{bmatrix}_{N_z \times N_z} \right)^{-1} \left( \underset{N_z \times N_u}{\left.\frac{\partial Z}{\partial u}\right|_{op}} + \begin{bmatrix} \underset{1 \times N_p}{\Delta p^T} \left.\frac{\partial^2 Z_1}{\partial p \partial u}\right|_{op} \\ \vdots \\ \underset{1 \times N_p}{\Delta p^T} \left.\frac{\partial^2 Z_{N_z}}{\partial p \partial u}\right|_{op} \end{bmatrix}_{N_z \times N_u} \right) \right] \underset{N_u \times 1}{\Delta u}
$$

$$
\underset{B(\Delta p) \atop N_x \times N_u}{}
$$

$$
+ \left[ \underset{N_x \times N_p}{\left.\frac{\partial X}{\partial p}\right|_{op}} + \frac{1}{2} \begin{bmatrix} \underset{1 \times N_p}{\Delta p^T} \left.\frac{\partial^2 X_1}{\partial p^2}\right|_{op} \\ \underset{N_p \times N_p}{} \\ \underset{1 \times N_p}{\Delta p^T} \left.\frac{\partial^2 X_2}{\partial p^2}\right|_{op} \\ \vdots \\ \underset{1 \times N_p}{\Delta p^T} \left.\frac{\partial^2 X_{N_x}}{\partial p^2}\right|_{op} \end{bmatrix}_{N_x \times N_p} - \left( \underset{N_x \times N_z}{\left.\frac{\partial X}{\partial z}\right|_{op}} + \begin{bmatrix} \underset{1 \times N_p}{\Delta p^T} \left.\frac{\partial^2 X_1}{\partial p \partial z}\right|_{op} \\ \underset{1 \times N_p}{\Delta p^T} \left.\frac{\partial^2 X_2}{\partial p \partial z}\right|_{op} \\ \vdots \\ \underset{1 \times N_p}{\Delta p^T} \left.\frac{\partial^2 X_{N_x}}{\partial p \partial z}\right|_{op} \end{bmatrix}_{N_x \times N_z} \right) \left( \underset{N_z \times N_z}{\left.\frac{\partial Z}{\partial z}\right|_{op}} + \begin{bmatrix} \underset{1 \times N_p}{\Delta p^T} \left.\frac{\partial^2 Z_1}{\partial p \partial z}\right|_{op} \\ \underset{1 \times N_p}{\Delta p^T} \left.\frac{\partial^2 Z_2}{\partial p \partial z}\right|_{op} \\ \vdots \\ \underset{1 \times N_p}{\Delta p^T} \left.\frac{\partial^2 Z_{N_z}}{\partial p \partial z}\right|_{op} \end{bmatrix}_{N_z \times N_z} \right)^{-1} \left( \underset{N_z \times N_p}{\left.\frac{\partial Z}{\partial p}\right|_{op}} + \frac{1}{2} \begin{bmatrix} \underset{1 \times N_p}{\Delta p^T} \left.\frac{\partial^2 Z_1}{\partial p^2}\right|_{op} \\ \underset{1 \times N_p}{\Delta p^T} \left.\frac{\partial^2 Z_2}{\partial p^2}\right|_{op} \\ \vdots \\ \underset{1 \times N_p}{\Delta p^T} \left.\frac{\partial^2 Z_{N_z}}{\partial p^2}\right|_{op} \end{bmatrix}_{N_z \times N_p} \right) \right] \underset{N_p \times 1}{\Delta p} \quad (13a)
$$

$$
\underset{X_p(\Delta p) \atop N_x \times N_p}{}
$$

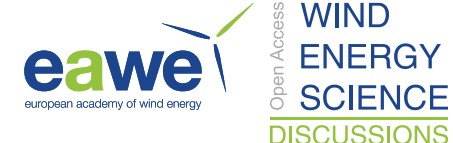

$$
\Delta y_{N_y \times 1} = \underbrace{\left[ \left.\frac{\partial Y}{\partial x}\right|_{op} + \begin{bmatrix} \Delta p^T \left.\frac{\partial^2 Y_1}{\partial p \partial x}\right|_{op} \\ \Delta p^T \left.\frac{\partial^2 Y_2}{\partial p \partial x}\right|_{op} \\ \vdots \\ \Delta p^T \left.\frac{\partial^2 Y_{N_y}}{\partial p \partial x}\right|_{op} \end{bmatrix} - \left[ \left.\frac{\partial Y}{\partial z}\right|_{op} + \begin{bmatrix} \Delta p^T \left.\frac{\partial^2 Y_1}{\partial p \partial z}\right|_{op} \\ \Delta p^T \left.\frac{\partial^2 Y_2}{\partial p \partial z}\right|_{op} \\ \vdots \\ \Delta p^T \left.\frac{\partial^2 Y_{N_y}}{\partial p \partial z}\right|_{op} \end{bmatrix}\right] \left[ \left.\frac{\partial Z}{\partial z}\right|_{op} + \begin{bmatrix} \Delta p^T \left.\frac{\partial^2 Z_1}{\partial p \partial z}\right|_{op} \\ \Delta p^T \left.\frac{\partial^2 Z_2}{\partial p \partial z}\right|_{op} \\ \vdots \\ \Delta p^T \left.\frac{\partial^2 Z_{N_z}}{\partial p \partial z}\right|_{op} \end{bmatrix}\right]^{-1} \left[ \left.\frac{\partial Z}{\partial x}\right|_{op} + \begin{bmatrix} \Delta p^T \left.\frac{\partial^2 Z_1}{\partial p \partial x}\right|_{op} \\ \Delta p^T \left.\frac{\partial^2 Z_2}{\partial p \partial x}\right|_{op} \\ \vdots \\ \Delta p^T \left.\frac{\partial^2 Z_{N_z}}{\partial p \partial x}\right|_{op} \end{bmatrix}\right]\right]}_{C(\Delta p) \;\; N_y \times N_x} \Delta x_{N_x \times 1}
$$

$$
+ \underbrace{\left[ \left.\frac{\partial Y}{\partial u}\right|_{op} + \begin{bmatrix} \Delta p^T \left.\frac{\partial^2 Y_1}{\partial p \partial u}\right|_{op} \\ \Delta p^T \left.\frac{\partial^2 Y_2}{\partial p \partial u}\right|_{op} \\ \vdots \\ \Delta p^T \left.\frac{\partial^2 Y_{N_y}}{\partial p \partial u}\right|_{op} \end{bmatrix} - \left[ \left.\frac{\partial Y}{\partial z}\right|_{op} + \begin{bmatrix} \Delta p^T \left.\frac{\partial^2 Y_1}{\partial p \partial z}\right|_{op} \\ \Delta p^T \left.\frac{\partial^2 Y_2}{\partial p \partial z}\right|_{op} \\ \vdots \\ \Delta p^T \left.\frac{\partial^2 Y_{N_y}}{\partial p \partial z}\right|_{op} \end{bmatrix}\right] \left[ \left.\frac{\partial Z}{\partial z}\right|_{op} + \begin{bmatrix} \Delta p^T \left.\frac{\partial^2 Z_1}{\partial p \partial z}\right|_{op} \\ \Delta p^T \left.\frac{\partial^2 Z_2}{\partial p \partial z}\right|_{op} \\ \vdots \\ \Delta p^T \left.\frac{\partial^2 Z_{N_z}}{\partial p \partial z}\right|_{op} \end{bmatrix}\right]^{-1} \left[ \left.\frac{\partial Z}{\partial u}\right|_{op} + \begin{bmatrix} \Delta p^T \left.\frac{\partial^2 Z_1}{\partial p \partial u}\right|_{op} \\ \Delta p^T \left.\frac{\partial^2 Z_2}{\partial p \partial u}\right|_{op} \\ \vdots \\ \Delta p^T \left.\frac{\partial^2 Z_{N_z}}{\partial p \partial u}\right|_{op} \end{bmatrix}\right]\right]}_{D(\Delta p) \;\; N_y \times N_u} \Delta u_{N_u \times 1}
$$

$$
+ \underbrace{\left[ \left.\frac{\partial Y}{\partial p}\right|_{op} + \frac{1}{2}\begin{bmatrix} \Delta p^T \left.\frac{\partial^2 Y_1}{\partial p^2}\right|_{op} \\ \Delta p^T \left.\frac{\partial^2 Y_2}{\partial p^2}\right|_{op} \\ \vdots \\ \Delta p^T \left.\frac{\partial^2 Y_{N_y}}{\partial p^2}\right|_{op} \end{bmatrix} - \left[ \left.\frac{\partial Y}{\partial z}\right|_{op} + \begin{bmatrix} \Delta p^T \left.\frac{\partial^2 Y_1}{\partial p \partial z}\right|_{op} \\ \Delta p^T \left.\frac{\partial^2 Y_2}{\partial p \partial z}\right|_{op} \\ \vdots \\ \Delta p^T \left.\frac{\partial^2 Y_{N_y}}{\partial p \partial z}\right|_{op} \end{bmatrix}\right] \left[ \left.\frac{\partial Z}{\partial z}\right|_{op} + \begin{bmatrix} \Delta p^T \left.\frac{\partial^2 Z_1}{\partial p \partial z}\right|_{op} \\ \Delta p^T \left.\frac{\partial^2 Z_2}{\partial p \partial z}\right|_{op} \\ \vdots \\ \Delta p^T \left.\frac{\partial^2 Z_{N_z}}{\partial p \partial z}\right|_{op} \end{bmatrix}\right]^{-1} \left[ \left.\frac{\partial Z}{\partial p}\right|_{op} + \frac{1}{2}\begin{bmatrix} \Delta p^T \left.\frac{\partial^2 Z_1}{\partial p^2}\right|_{op} \\ \Delta p^T \left.\frac{\partial^2 Z_2}{\partial p^2}\right|_{op} \\ \vdots \\ \Delta p^T \left.\frac{\partial^2 Z_{N_z}}{\partial p^2}\right|_{op} \end{bmatrix}\right]\right]}_{Y_p(\Delta p) \;\; N_y \times N_p} \Delta p_{N_p \times 1}
$$

(13b)



$$\underset{N_z\times1}{\Delta z} = -\left[\underset{\substack{N_z\times N_z}}{\left.\frac{\partial Z}{\partial z}\right|_{op}} + \begin{bmatrix} \underset{1\times N_p}{\Delta p^T} \left.\frac{\partial^2 Z_1}{\partial p \partial z}\right|_{op} \\ \underset{N_p\times N_z}{} \\ \underset{1\times N_p}{\Delta p^T} \left.\frac{\partial^2 Z_2}{\partial p \partial z}\right|_{op} \\ \underset{N_p\times N_z}{} \\ \vdots \\ \underset{1\times N_p}{\Delta p^T} \left.\frac{\partial^2 Z_{N_z}}{\partial p \partial z}\right|_{op} \\ \underset{N_p\times N_z}{} \end{bmatrix}\right]^{-1} \left\{ \left[\underset{\substack{N_z\times N_x}}{\left.\frac{\partial Z}{\partial x}\right|_{op}} + \begin{bmatrix} \underset{1\times N_p}{\Delta p^T} \left.\frac{\partial^2 Z_1}{\partial p \partial x}\right|_{op} \\ \underset{N_p\times N_x}{} \\ \underset{1\times N_p}{\Delta p^T} \left.\frac{\partial^2 Z_2}{\partial p \partial x}\right|_{op} \\ \underset{N_p\times N_x}{} \\ \vdots \\ \underset{1\times N_p}{\Delta p^T} \left.\frac{\partial^2 Z_{N_z}}{\partial p \partial x}\right|_{op} \\ \underset{N_p\times N_x}{} \end{bmatrix}\right] \underset{N_x\times1}{\Delta x} + \left[\underset{\substack{N_z\times N_u}}{\left.\frac{\partial Z}{\partial u}\right|_{op}} + \begin{bmatrix} \underset{1\times N_p}{\Delta p^T} \left.\frac{\partial^2 Z_1}{\partial p \partial u}\right|_{op} \\ \underset{N_p\times N_u}{} \\ \underset{1\times N_p}{\Delta p^T} \left.\frac{\partial^2 Z_2}{\partial p \partial u}\right|_{op} \\ \underset{N_p\times N_u}{} \\ \vdots \\ \underset{1\times N_p}{\Delta p^T} \left.\frac{\partial^2 Z_{N_z}}{\partial p \partial u}\right|_{op} \\ \underset{N_p\times N_u}{} \end{bmatrix}\right] \underset{N_u\times1}{\Delta u} + \left[\underset{\substack{N_z\times N_p}}{\left.\frac{\partial Z}{\partial p}\right|_{op}} + \frac{1}{2}\begin{bmatrix} \underset{1\times N_p}{\Delta p^T} \left.\frac{\partial^2 Z_1}{\partial p^2}\right|_{op} \\ \underset{N_p\times N_p}{} \\ \underset{1\times N_p}{\Delta p^T} \left.\frac{\partial^2 Z_2}{\partial p^2}\right|_{op} \\ \underset{N_p\times N_p}{} \\ \vdots \\ \underset{1\times N_p}{\Delta p^T} \left.\frac{\partial^2 Z_{N_z}}{\partial p^2}\right|_{op} \\ \underset{N_p\times N_p}{} \end{bmatrix}\right] \underset{N_p\times1}{\Delta p}\right\} \tag{14}$$

## 2.4 Observations

The main physical insights that can be obtained by reviewing these mathematical details are summarized as follows:

1) The Hessians represent the change in Jacobians associated with the parameter variations and many Hessians are needed to represent the parameterized linear state-space matrices; that said, while the Hessian matrices could be fully populated—see Eq. (6)—they are likely quite sparse in practice;

2) The constraints, if present in the underlying nonlinear model, cannot be algebraically eliminated until the parameter perturbations are explicitly set; this means that while the Jacobians and Hessians can be computed based only on knowledge of the parameter operating point, much of the algebraic manipulation to define the parameterized linear state-space matrices must be implemented in a post-processing step (once the parameter perturbations are explicitly set);

3) The parameterized linear state-space matrices are likely only valid for small parameter perturbations; for large parameter variations, multiple parameter operating points would need to be defined, $\left.p\right|_{op}$, and new Jacobians and Hessians would have to be computed for each parameter operating point;

4) Parameter variations result in changes to the continuous-state and output operating points, $\left.\Delta x\right|_{op}$ and $\left.\Delta y\right|_{op}$;

5) Within the parameterized linear state-space matrices, the variation with parameter is linear by design, except when constraint states exist, whereby the variation with parameter may include nonlinear relations of the parameter perturbations (as a result of the matrix inverse inherent in Eq. (14));

6) Within the operating point changes associated with the parameter variations, $\left.\Delta x\right|_{op}$ and $\left.\Delta y\right|_{op}$, there are nonlinear relations of the parameter perturbations (as a result of the matrix inverse inherent in Eq. (10));



7)  The Hessians with respect to the parameters only, $\dfrac{\partial^2}{\partial p^2}$, are only needed to improve the evaluation of the constraints

and the change in operating points associated with the parameter variations; they do not affect the resulting parameterized linearized state-space matrices, and so, could be neglected if the linearized state-space matrices are of more interest than the change in operating points or constraints; and

8)  When $\Delta p$ equals zero, the parameterized linear state-space model from Eqs. (11) and (12) reduces to the original linearized model in Eqs. (2) and (3), as expected.

The first three items in this summary deterred us from implementing the theoretical approach outlined here directly within OpenFAST.

The same approach applied above can be used to find the linearization with respect to design parameters of the overall coupled nonlinear system across all modules of OpenFAST—the parameterized extension of Eq. (18) from Jonkman, 2013—but this extension does not provide any new insight and is not shown here.

**3 Illustrative Mass-Spring-Damper Example**

**3.1 System and linearization without parameters**

To illustrate the theory developed in Sect. 2, the equations are applied to a simple forced mass-spring-damper system. Figure 1 visualizes the system, where $m$ is the mass, $c$ is the damping of the dashpot, $k$ is the stiffness of the spring, $q$ is the displacement of the mass, $F$ is the force applied in the direction of displacement, and $g$ is the gravitational acceleration.

The first-order system of Eq. (1) can be established by defining the states, inputs, outputs, and parameters as in Eq. (15). For illustrative purposes, the applied force is characterized as an input and the output is arbitrarily characterized as the full motion of the mass (displacement, $q$, velocity, $\dot{q}$, and acceleration, $\ddot{q}$), as well as the force transmitted to the foundation, $F_{Transmitted}$. There are no constraint (algebraic) states in this system, so, $z$ and $Z(\ )$ are empty, represented by $\varnothing$, and are neglected from subsequent equations. All parameters are included in $p$ for illustrative purposes, although gravity would not typically

be a design variable.



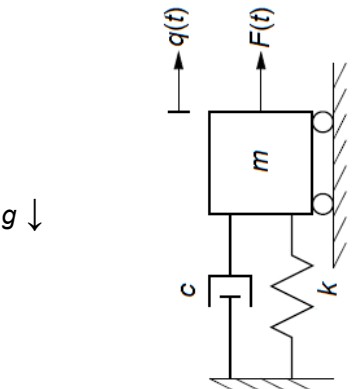

**Figure 1: Visualization of the forced mass-spring-damper system.**

$$x = \begin{Bmatrix} q \\ \dot{q} \end{Bmatrix}, \; \dot{x} = \begin{Bmatrix} \dot{q} \\ \ddot{q} \end{Bmatrix}, \; z = \varnothing, \; u = F, \; y = \begin{Bmatrix} q \\ \dot{q} \\ \ddot{q} \\ F_{Transmitted} \end{Bmatrix}, \; p = \begin{Bmatrix} m \\ c \\ k \\ g \end{Bmatrix} \tag{15}$$

The first-order system of Eq. (1) for the simple forced mass-spring-damper system can then be written following Newton's
second law, as in Eq. (16).

$$\dot{x} = X(x, u, p) = \begin{bmatrix} 0 & 1 \\ -\dfrac{k}{m} & -\dfrac{c}{m} \end{bmatrix} x + \begin{Bmatrix} 0 \\ \dfrac{1}{m} \end{Bmatrix} u + \begin{Bmatrix} 0 \\ -g \end{Bmatrix}$$

$$y = Y(x, u, p) = \begin{bmatrix} 1 & 0 \\ 0 & 1 \\ -\dfrac{k}{m} & -\dfrac{c}{m} \\ k & c \end{bmatrix} x + \begin{Bmatrix} 0 \\ 0 \\ \dfrac{1}{m} \\ 0 \end{Bmatrix} u + \begin{Bmatrix} 0 \\ 0 \\ -g \\ 0 \end{Bmatrix} \tag{16}$$

The operating point in this is example is taken to be the static equilibrium of the mass-spring-damper system in absence of
external forcing (input), as given in Eq. (17). The simple forced mass-spring-damper system is already linear in nature, so, Eq.
(3) with fixed parameters can be written directly, as in Eq. (18).





$$x\Big|_{op} = \left\{ \begin{matrix} -\dfrac{mg}{k} \\ 0 \end{matrix} \right\}, \ \dot{x}\Big|_{op} = \left\{ \begin{matrix} 0 \\ 0 \end{matrix} \right\}, \ u\Big|_{op} = 0, \ y\Big|_{op} = \left\{ \begin{matrix} -\dfrac{mg}{k} \\ 0 \\ 0 \\ -mg \end{matrix} \right\}, \ p = \left\{ \begin{matrix} m \\ c \\ k \\ g \end{matrix} \right\} \tag{17}$$

$$\Delta\dot{x} = \underbrace{\begin{bmatrix} 0 & 1 \\ -\dfrac{k}{m} & -\dfrac{c}{m} \end{bmatrix}}_{A=\frac{\partial X}{\partial x}\big|_{op}} \Delta x + \underbrace{\left\{ \begin{matrix} 0 \\ \dfrac{1}{m} \end{matrix} \right\}}_{B=\frac{\partial X}{\partial u}\big|_{op}} \Delta u \tag{18}$$

$$\Delta y = \underbrace{\begin{bmatrix} 1 & 0 \\ 0 & 1 \\ -\dfrac{k}{m} & -\dfrac{c}{m} \\ k & c \end{bmatrix}}_{C=\frac{\partial Y}{\partial x}\big|_{op}} \Delta x + \underbrace{\left\{ \begin{matrix} 0 \\ 0 \\ \dfrac{1}{m} \\ 0 \end{matrix} \right\}}_{D=\frac{\partial Y}{\partial u}\big|_{op}} \Delta u$$

### 3.2 Analytical linearization with respect to parameters

Now we wish to understand the impact of small deviations in the parameters on the linear state-space model. Of course, because
the linear state-space model and associated operating point have already been expressed analytically in Eqs. (17) and (18)
(which is not typically the case in complex wind turbine dynamics models like OpenFAST), it is trivial to write down what
the parameter-dependent variations of the linear state matrices and operating point changes should be exactly, as shown in
Eqs. (19) and (20).





$$
\begin{aligned}
A(\Delta p)\Big|_{exact} &= \begin{bmatrix} 0 & 1 \\ -\dfrac{k+\Delta k}{m+\Delta m} & -\dfrac{c+\Delta c}{m+\Delta m} \end{bmatrix} \\[2mm]
B(\Delta p)\Big|_{exact} &= \left\{ \begin{array}{c} 0 \\ \dfrac{1}{m+\Delta m} \end{array} \right\} \\[2mm]
C(\Delta p)\Big|_{exact} &= \begin{bmatrix} 1 & 0 \\ 0 & 1 \\ -\dfrac{k+\Delta k}{m+\Delta m} & -\dfrac{c+\Delta c}{m+\Delta m} \\ k+\Delta k & c+\Delta c \end{bmatrix} \\[2mm]
D(\Delta p)\Big|_{exact} &= \left\{ \begin{array}{c} 0 \\ 0 \\ \dfrac{1}{m+\Delta m} \\ 0 \end{array} \right\}
\end{aligned}
\tag{19}
$$

$$
\Delta x\big|_{op\ exact} = \left\{ \begin{array}{c} -\dfrac{(m+\Delta m)(g+\Delta g)}{(k+\Delta k)} \\ 0 \end{array} \right\} - \left\{ \begin{array}{c} -\dfrac{mg}{k} \\ 0 \end{array} \right\} = \left\{ \begin{array}{c} -\dfrac{km\Delta g + k\Delta mg + k\Delta m\Delta g - \Delta kmg}{k(k+\Delta k)} \\ 0 \end{array} \right\}
\tag{20}
$$

$$
\Delta y\big|_{op\ exact} = \left\{ \begin{array}{c} -\dfrac{(m+\Delta m)(g+\Delta g)}{(k+\Delta k)} \\ 0 \\ 0 \\ -(m+\Delta m)(g+\Delta g) \end{array} \right\} - \left\{ \begin{array}{c} -\dfrac{mg}{k} \\ 0 \\ 0 \\ -mg \end{array} \right\} = \left\{ \begin{array}{c} -\dfrac{km\Delta g + k\Delta mg + k\Delta m\Delta g - \Delta kmg}{k(k+\Delta k)} \\ 0 \\ 0 \\ -\Delta mg - m\Delta g - \Delta m\Delta g \end{array} \right\}
$$

### 3.3 Hessian-based linearization with respect to parameters

Now we wish to understand the impact of small deviations in the parameters on the linear state-space model following the theory presented in Sect. 2. In this simple example, the Hessians can be computed analytically. An example Hessian of the continuous-state functions with respect to parameters and continuous states evaluated at the operating point—written generically in Eq. (6)—is given for this simple forced mass-spring-damper example in Eq. (21). Note that in this example, $N_x = 2$, $N_z = 0$, $N_u = 1$, $N_y = 4$, and $N_p = 4$.



$$\frac{\partial^2 X_1}{\partial p \partial x}\bigg|_{op} = \begin{bmatrix} 0 & 0 \\ 0 & 0 \\ 0 & 0 \\ 0 & 0 \end{bmatrix}, \quad \frac{\partial^2 X_2}{\partial p \partial x}\bigg|_{op} = \begin{bmatrix} \dfrac{k}{m^2} & \dfrac{c}{m^2} \\ 0 & -\dfrac{1}{m} \\ -\dfrac{1}{m} & 0 \\ 0 & 0 \end{bmatrix} \tag{21}$$

Carrying out the remainder of the math, Eq. (22) provides the parameter-dependent variations of the linear state matrices from Eq. (8) and Eq. (23) provides the change in operating point from Eq. (10) for this simple forced mass-spring damper example.

$$A(\Delta p) = \begin{bmatrix} 0 & 1 \\ \left(\dfrac{k\Delta m}{m^2} - \dfrac{(k+\Delta k)}{m}\right) & \left(\dfrac{c\Delta m}{m^2} - \dfrac{(c+\Delta c)}{m}\right) \end{bmatrix}$$

$$B(\Delta p) = \left\{ \begin{matrix} 0 \\ \dfrac{1}{m} - \dfrac{\Delta m}{m^2} \end{matrix} \right\}$$

$$C(\Delta p) = \begin{bmatrix} 1 & 0 \\ 0 & 1 \\ \left(\dfrac{k\Delta m}{m^2} - \dfrac{(k+\Delta k)}{m}\right) & \left(\dfrac{c\Delta m}{m^2} - \dfrac{(c+\Delta c)}{m}\right) \\ k + \Delta k & c + \Delta c \end{bmatrix}$$

$$D(\Delta p) = \left\{ \begin{matrix} 0 \\ 0 \\ \dfrac{1}{m} - \dfrac{\Delta m}{m^2} \\ 0 \end{matrix} \right\} \tag{22}$$

$$\Delta x\big|_{op} = \left\{ \begin{matrix} -\dfrac{km\Delta mg - k\Delta m^2 g + \Delta km\Delta mg - \Delta km^2 g + km^2\Delta g}{k(k+\Delta k)m - k^2\Delta m} \\ 0 \end{matrix} \right\} \tag{23}$$

$$\Delta y\big|_{op} = \left\{ \begin{matrix} -\dfrac{km\Delta mg - k\Delta m^2 g + \Delta km\Delta mg - \Delta km^2 g + km^2\Delta g}{k(k+\Delta k)m - k^2\Delta m} \\ 0 \\ 0 \\ -\dfrac{k^2 m\Delta mg + k(k+\Delta k)m^2\Delta g - k(k+\Delta k)\Delta m^2 g + \Delta k(k+\Delta k)m\Delta mg}{k(k+\Delta k)m - k^2\Delta m} \end{matrix} \right\}$$




### 3.4 Observations

Comparing the two formulations—Eq. (22) with Eq. (19) and Eq. (23) with Eq. (20)—the following physical insights can be summarized:

- In the Hessian-based formulation of the parameter-dependent variations of the linear state matrices, all parameter perturbations appear linear in nature because this simple example does not have constraint states. This follows directly from item five in Sect. 2. The end result is that any parameters that are linear in nature in the underlying formulation (e.g., damping, $c$, and stiffness, $k$) are expressed exactly in the Hessian-based formulation;

- The mass, $m$, however, is nonlinear in the underlying formulation (showing up as $\dfrac{1}{m}$), and so, in the Hessian-based

formulation of the parameter-dependent variations of the linear state matrices, the mass effect is approximated. Effectively, $\dfrac{1}{m+\Delta m}$ in the exact formulation has been approximated as $\dfrac{1}{m}-\dfrac{\Delta m}{m^2}$ in the Hessian-based formulation, and equating too, we see that $m^2 \approx (m-\Delta m)(m+\Delta m) = m^2 - \Delta m^2$ only holds true when $\Delta m^2 \ll m^2$, which is a second-order error. The end result is that any parameters that are nonlinear in nature in the underlying formulation are approximated to first-order accuracy in the Hessian-based formulation of the parameter-

dependent variations of the linear state matrices;

- Within the operating point changes associated with the parameter variations, there are nonlinear relations of the parameter perturbations (as a result of the matrix inverse inherent in Eq. (10)), which follows directly from item six in Sect. 2. However, the operating point changes are still not entirely exact in the Hessian-based formulation. It is interesting to note that the operating point changes are exact when any given parameter variation is treated in isolation

(with others zeroed), e.g., $\Delta q\big|_{op} = -\dfrac{\Delta mg}{k}$ and $\Delta F_{Transmission}\big|_{op} = -\Delta mg$ when $\Delta c = \Delta k = \Delta g = 0$; likewise for other one-off parameter variations; and

- When Hessians with respect to the parameters only, $\dfrac{\partial^2}{\partial p^2}$, are neglected as discussed in item seven of Sect. 2, the Hessian-based operating point changes from Eq. (23) simplify a bit, as shown in Eq. (24). In Eq. (24), the operating point changes are still exact for one-off parameter variations of damping, $c$, and stiffness, $k$, and gravity, $g$, but

the operating point changes are no longer exact when $\Delta c = \Delta k = \Delta g = 0$, i.e., $\Delta q\big|_{op} = -\dfrac{m\Delta mg}{k(m-\Delta m)}$ and



$\Delta F_{Transmission}\big|_{op} = -\dfrac{m\Delta mg}{m-\Delta m}$ . This demonstrates that the inclusion of the $\dfrac{\partial^2}{\partial p^2}$ terms in the Hessians improves the change in operating points.

$$\Delta x\big|_{op \; neglecting \frac{\partial}{\partial p^2}} = \left\{ \begin{array}{c} -\dfrac{km\Delta mg - \Delta km^2 g + km^2\Delta g}{k\left(k+\Delta k\right)m - k^2\Delta m} \\ 0 \end{array} \right\}$$

$$\Delta y\big|_{op \; neglecting \frac{\partial}{\partial p^2}} = \left\{ \begin{array}{c} -\dfrac{km\Delta mg - \Delta km^2 g + km^2\Delta g}{k\left(k+\Delta k\right)m - k^2\Delta m} \\ 0 \\ 0 \\ -\dfrac{km\Delta mg + \left(k+\Delta k\right)m^2\Delta g}{\left(k+\Delta k\right)m - k\Delta m} \end{array} \right\} \tag{24}$$

## 4 Numerical Implementation in WEIS

### 4.1 The WEIS framework

While the theoretical development presented in Sect. 2 has not been implemented in OpenFAST, we implemented the direct evaluation and interpolation methods summarized in Sect. 1 within WEIS. The primary goal of WEIS is to provide a framework for the CCD of a floating wind turbine controller alongside turbine and platform geometry at multiple fidelity levels. The advantage of using WEIS for this work is that the wind turbine system is controlled by high-level design variables where each

variable can affect several OpenFAST inputs at once. For instance, a change of tower material mass density will affect a change of the distributed properties of the tower but also a change of the shape functions of the tower, both of which are inputs to OpenFAST. WEIS handles such interdependencies automatically by propagating the effects of design variable changes through the entire wind turbine system. In the following subsection, we detail the how the interpolation and direct evaluation methods were implemented in WEIS. While all parameterized linear state-space matrices can be obtained following this

implementation, only the parameterized state matrix, $A(\Delta p)$, is used the numerical case study of Sect. 5 and discussed below.

OpenFAST is used within WEIS but was not modified in any way for this work.

### 4.2 Design of experiments

For this paper, the WEIS framework was adapted to incorporate the following workflow: call OpenFAST, retrieve the linearized state-space matrices, and store them at each evaluation call. In this study, the evaluations are not done within an

optimization loop, but within a parametric loop, also referred to as design of experiment or parameter sweep. We extended



WEIS to be able to run design of experiments where: the user specifies a set of $N_p$ design variables, $p_{n_p}$ for $n_p = \{1, 2, \ldots, N_p\}$, the interval over which the variables are to be varied, $\left[ p_{n_p, \min} ; p_{n_p, \max} \right]$, and the number of subdivisions of each interval (linear spacing is used). Based on these user-specified settings, WEIS evaluates for all the combinations of the design variable values. In this paper, the term parameter is used in place of design variables.

### 4.3 Implementation of the direct evaluation method

No specific treatment is needed for the direct evaluation method. For each parameter point, $p = p\big|_{op} + \Delta p$ ( $p\big|_{op}$ is explicitly defined in the next section), the linearized state matrix, $A(\Delta p)$, is evaluated using a call to the linearization functionality of OpenFAST. The time necessary for a direct call is in the order of minutes for onshore wind turbines and up to an hour for a complex floating offshore wind turbine. The linearization time itself only accounts for about 1 minute of this total, but OpenFAST currently establishes the operating point by performing a trim solution using a time marching loop. Due to the low frequencies of floating platforms, a significant amount of simulation time is necessary for the system to reach an equilibrium. In a separate project, we are currently working on a direct steady-state solve within OpenFAST to avoid this trim solution, which will significantly reduce the computational time needed.

### 4.4 Implementation of the interpolation method

The implementation of the interpolation method requires two WEIS steps: a pre-processing step, and a replacement of the OpenFAST call with an interpolation call. The pre-processing step, proceeds as follows:

- A nominal parameter point (operating point) is defined at the centre of all the parameter intervals:

$$p_{n_p}\big|_{op} = \frac{p_{n_p, \max} + p_{n_p, \min}}{2} \text{ for } n_p = \{1, 2, \ldots, N_p\}, \tag{25}$$

- The nominal linearized state matrix is obtained using an OpenFAST call at this operating point, $A$,

- For each parameter, of index $n_p$, two OpenFAST evaluations are done to retrieve the state-space model at the bounds of the parameter interval, $A\left( \delta p_{n_p} e_{n_p} \right)$, and $A\left( -\delta p_{n_p} e_{n_p} \right)$, where the parameter variation for each parameter, $\delta p_{n_p}$, is defined in Eq. (26) and $e_{n_p}$ is a unit vector the same size of $p$ with zeros for each element except for index $n_p$, which equals unity.

$$\delta p_{n_p} = \frac{p_{n_p, \max} - p_{n_p, \min}}{2} \text{ for } n_p = \{1, 2, \ldots, N_p\}. \tag{26}$$



• The slopes, $\left[\dfrac{dA}{dp}\right]_{n_p}$ , corresponding to each parameter variation, each of which is a matrix, are then computed using

a central finite difference:

$$\left[\frac{dA}{dp}\right]_{n_p} = \frac{A\left(\delta p_{n_p} e_{n_p}\right) - A\left(-\delta p_{n_p} e_{n_p}\right)}{2\delta p_{n_p}} \text{ for } n_p = \left\{1, 2, \ldots, N_p\right\}. \tag{27}$$

In total, the pre-processing steps consists of $2N_p + 1$ direct evaluations (calls to the OpenFAST linearization functionality).

The nominal state-space model and slopes are stored for later use in the interpolation step. After the pre-processing step, the

WEIS optimization loop, or design of experiment loop, proceed as usual, but whenever a linearized state-space model is needed

for a given change in operating point, $\Delta p$ , instead of calling OpenFAST, an interpolation is done according to Eq. (28). The

evaluation time is in the order of milliseconds, which is significant smaller than a direct evaluation call.

$$A(\Delta p) = A + \sum_{n_p = 1}^{N_p} \Delta p_{n_p} \left[\frac{dA}{dp}\right]_{n_p} \tag{28}$$

Figure 2 illustrates the interpolation method for the case with two parameters variations ( $N_p = 2$ ).

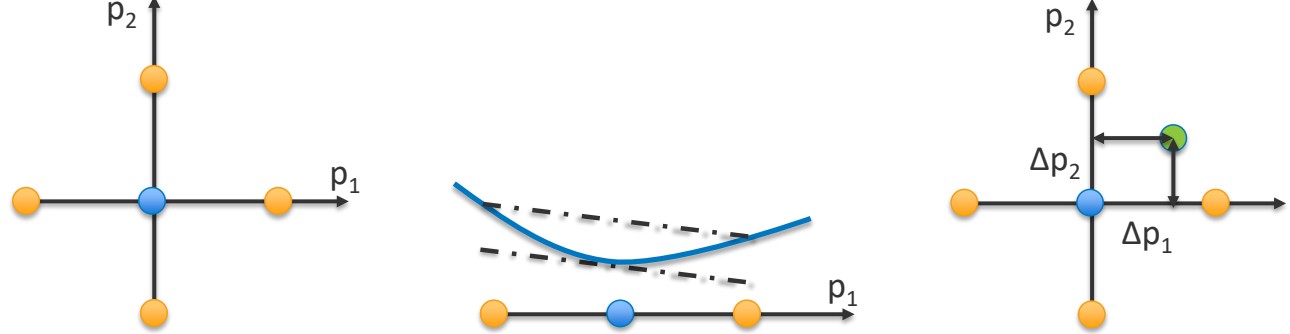


**Figure 2: Visualization of the interpolation method for the case with two parameter variations. Left: Nominal (operating) parameter point in blue and the bounds of the parameter interval in orange. Middle: Slope (shown as a the dash-dot line) of the state-space matrix values (shown as a continuous blue function) computed using finite differences from the bounds. Right: interpolation step.**

## 5 Numerical Case Study Using WEIS and OpenFAST

In this section, we apply the numerical methods presented in Sect. 4 to the IEA 15-MW reference wind turbine (Gaertner et al., 2020), placed atop the University of Maine semisubmersible (Allen et al., 2020). For simplicity, the sway and roll degrees of freedom of the floater are disabled, but other relevant structural degrees of freedom are enabled. The wind turbine rotates at a constant rotor speed of 5 rpm with a blade pitch of 2.7 deg, corresponding to the operating conditions for a wind speed at hub height of 5 m/s and a power law exponent of 0.12. These conditions, corresponding to 10% of rated power, were chosen



to ease the identification of the turbine modes (the automatic identification of modes above rated can be difficult for highly flexible rotors). The turbine is modelled using the following modules of OpenFAST: InflowWind, AeroDyn, ElastoDyn, HydroDyn and MAP++. For the linearization, the aerodynamics are modelled using the Blade Element / Momentum model, with static airfoil data and frozen wake. The hydrodynamics of the platform are modelled with a hybrid combination of potential flow and a quadratic drag matrix, with the potential-flow solution in state-space form.

We chose to vary three different parameters: the tower mass density, varying between 5460 and 10140 kg/m$^3$; the tower Young's modulus, varying between 1.4e10 and 2.6e10 N/m$^2$; and the mooring line unstretched (rest) length, varying between 800 and 900 m. The variations of the tower properties correspond to ±30% of their nominal value, whereas the rest length is varied with ±6% due to the important non-linearity expected for this parameter. The nominal and parameterized linear state-space matrices are obtained using both the direct and interpolation methods for all combination of parameters, using nine

values per range. Based on the state matrix, an eigenvalue analysis and modal identification are performed for each set of parameters, and the results from both methods are compared in terms of the damped frequencies and linearized damping ratio for each full-system mode of the system. We present subset of results of this parametric study in the following paragraphs.

The results for one-dimensional variation of the tower properties are given in Figure 3 ($N_p = 1$). In this case, only the parameter on the abscissas are varied and other parameters are kept at their nominal values. A logarithmic scale is used on the

ordinates to better distinguish between the different modes. The following abbreviations are used in the figure for the tower modes: fore-aft (FA) bending, side-side (SS) bending. The interpolation method requires three evaluations of OpenFAST and the direct method requires nine evaluations of OpenFAST per parameter.

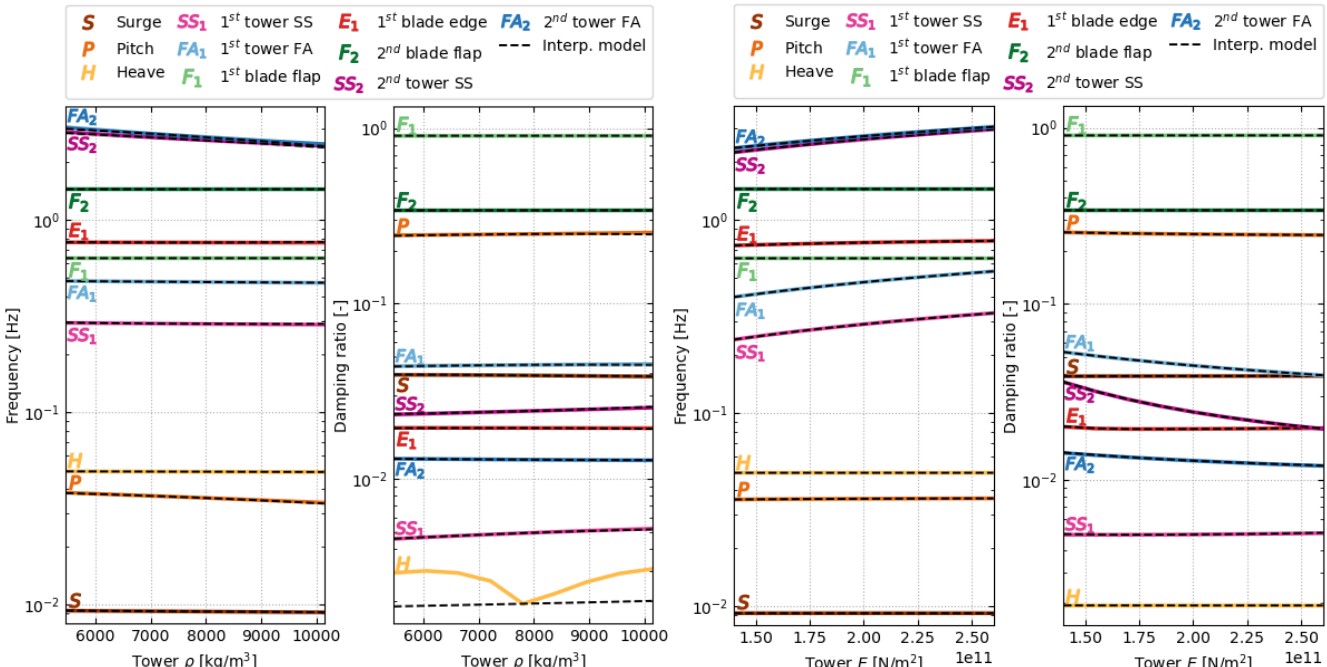

**Figure 3: Variation of the system damped frequencies and damping ratios as function of tower properties evaluated using the direct method (plain lines) and the interpolation method (dashed lines). Left: Variation of the tower density, $\rho$. Right: variation of the tower Young's modulus, $E$.**

From Figure 3, we observe that the interpolation method and the direct method are in strong agreement for all the modes of the structure. With the scale provided on the figure, the curves from both methods collapse on top of each other, except for the damping of the platform-heave mode when the tower density varies. We believe this is due to the numerical sensitivity of the eigenvalue analysis and the small value of the damping of this mode. The interpolation method is only expected to return the same value as the direct method when all values of the design parameters are at the nominal value. In the figure, this corresponds to the middle point of the abscissas. We observe that the variations of damped frequencies and damping ratios with the tower properties are well captured by the interpolation method, with the following expected behaviour: the frequencies of the tower modes decrease with increasing density and increase with increasing stiffness. We also note that, despite the linear characteristics assumed for the state matrix in the interpolation method, the frequencies and damping ratios display nonlinear behaviour after performing the eigenvalue analysis. A zoomed-in view of the variation of the second tower modes is given in Figure 4. With this axis scale, we can see some error between the interpolation and direct methods and that the error increases as the material density gets further away from the nominal value. No error is visible for variation of the material stiffness. This further exemplifies the results of Sect. 3: linearity is expected for variation of the stiffness but variations in mass are nonlinear and hence more difficult to capture by the interpolation method.

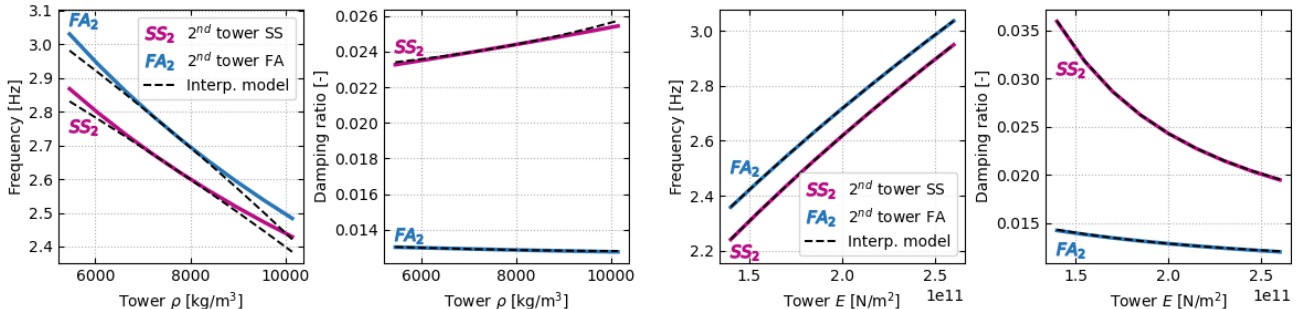

**Figure 4: Variation of the second tower mode damped frequencies and damping ratios as function of tower properties evaluated using the direct method (plain lines) and the interpolation method (dashed lines). Left: Variation of the tower density, $\rho$. Right: variation of the tower Young's modulus.**

We now present results where the tower density and stiffness are changed together ($N_p = 2$), using 9 values for each parameter, for a total of 81 evaluations. That is, the interpolation method requires five evaluations of OpenFAST and the direct method requires 81 evaluations of OpenFAST. The interpolated method is only expected to be exact at the centre of the parametric domain, that is, at the nominal values of the tower stiffness and density. The relative error in damped frequencies and damping ratios between the interpolated and direct methods is plotted in Figure 5 for the five modes that vary the most: the platform-pitch mode, and the two first tower-bending modes in the FA and SS directions.

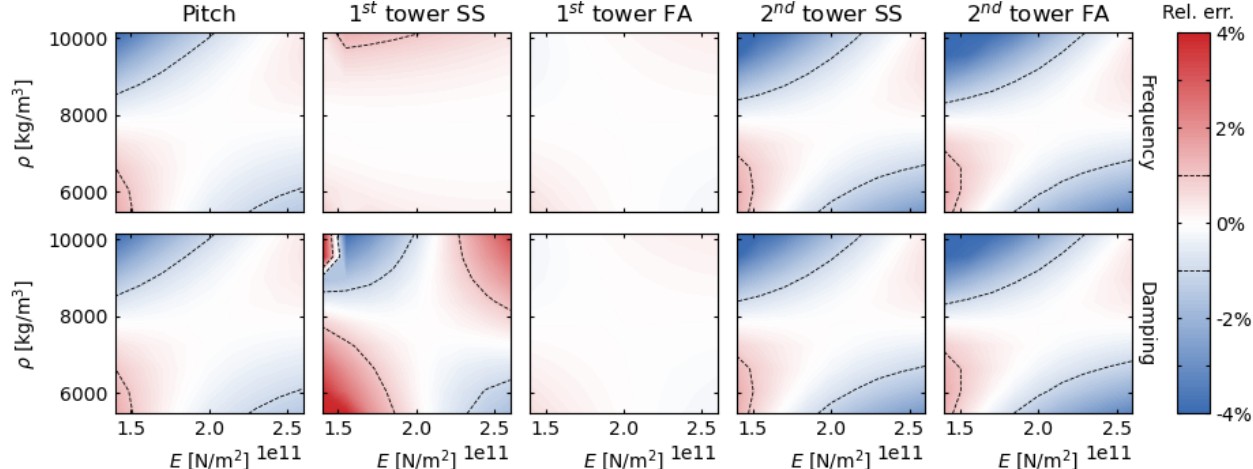

**Figure 5: Relative errors in the system frequencies and damping ratios between the direct and interpolation methods as function of tower properties (Young's modulus, *E*, and mass density, *ρ*). Top: Relative error in damped frequencies. Bottom: relative error in damping ratios.**

The results from Figure 5 confirm that the error of the interpolation method increases as the parameters get further away from the nominal values. The error is typically smaller along lines centred on the nominal values, and typically larger in the "corners". This is expected since the interpolation method uses slopes computed in each of the parameter directions but does not account for slopes in other (combined) directions. Overall, despite parameter variations of ±30% of their nominal value, the error of the interpolated method remains within ±5% for both damped frequencies and damping ratios.

In the final set of results, we present the impact of the mooring line unstretched length on the wind turbine modes as captured by both methods. Again, with $N_p = 1$, the interpolation method requires three evaluations of OpenFAST and the direct method requires nine evaluations of OpenFAST. As seen in Figure 6, the mooring length mostly affects the surge mode of the semi-submersible, with some effect on the pitch mode. The interpolation method captures most of the trends, but the error in the surge mode appears more significant than what was observed for the tower study. This is likely the result of the strong

nonlinear effect line length has on the force-displacement relationships of a catenary mooring system. Additionally, the surge mode becomes undetectable (and so is not plotted) using the state matrix obtained with the interpolation method for increased values of the mooring line unstretched length. It appears that the interpolation method fails in this situation, likely due to the numerical sensitivity of the eigenvalue analysis, i.e., the interpolation method does not provide enough precision for accurate eigensolution of all modes.





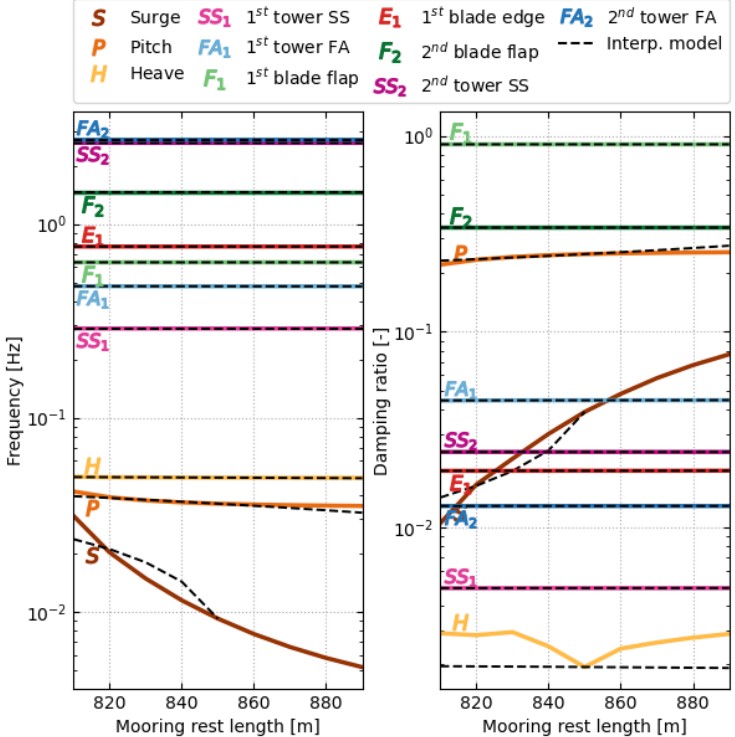


**Figure 6: Variation of the system damped frequencies and damping ratios as function of the mooring line unstretched length evaluated using the direct method (plain lines) and the interpolation method (dashed lines).**

## Conclusions

This work summarizes efforts done to understand the impact of design parameter variations in the physical system (mass,
stiffness, geometry, etc.) on the linearized system using OpenFAST. The results of this work have influenced the development
of the WEIS toolset being developed at NREL within the ATLANTIS program, whose goal is to enable CCD of the floating
wind turbine structure and controller.

Theoretical developments provide some physical insights, such as the impact of design parameter variations on the operation
point, the role of specific Hessians on specific state matrices and the operating point changes (including when algebraic
constraint states are present), and the errors that can arise when parameters function nonlinearly within the system. These
insights were further exemplified in a simple forced mass-spring-damper system example, but characteristics of the theoretical
approach deterred NREL from implementing it directly within OpenFAST. Instead, a direct evaluation method and an
interpolation method were implemented in the WEIS toolset, which makes use of OpenFAST, which were compared in terms
of computational cost and through a numerical case study. The results from the numerical examples were encouraging for the
tower study, whereby the interpolation method yielded damped frequency and damping ratio results close to the direct
evaluation method despite large variation in tower density and tower Young's modulus from their nominal values. However,

the interpolation method was less effective for the mooring line length study due to the strong nonlinear effect that line length has on the force-displacement relationships of a catenary mooring system. Moreover, the interpolation method failed for increased mooring rest length (with the surge mode undetectable), likely due to the numerical sensitivity of the eigenvalue
analysis.

Improvements to the interpolation method are possible but are left as future work. For example, an $N_p$-linear interpolation (e.g., bilinear interpolation for $N_p = 2$ or trilinear interpolation for $N_p = 3$) based on values at the corners of the $N_p$ hypercube could be used rather than using the centres of the faces of the hypercube, as is done here. This would require $2^{N_p} + 1$ evaluations of OpenFAST in the pre-processing step rather than $2N_p + 1$ evaluations used here, which would
improve the interpolation when multiple parameters are changed at the same time. Also, more advanced interpolations, such as $N_p$-cubic could be postulated.

Moreover, we may reconsider implementing the calculation of Hessians directly within OpenFAST, at least for some modules and classes of parameters.

**Author contribution**

Jason Jonkman developed the theoretical details and illustrative mass-spring-damper case study. Emmanuel Branlard developed the numerical case study. John Jasa implemented the numerical approach within WEIS.

**Competing interests**

The authors declare that they have no conflict of interest.

**Acknowledgements**

This work was authored by the National Renewable Energy Laboratory, operated by the Alliance for Sustainable Energy, LLC, for the U.S. Department of Energy (DOE) under Contract No. DE-AC36-08GO28308. Funding provided by the DOE ARPA-E ATLANTIS program under a 2020 project titled "WEIS: A Tool Set to Enable Controls Co-Design of Floating Offshore Wind Energy Systems." The views expressed in the article do not necessarily represent the views of the DOE or the U.S. Government. The U.S. Government retains and the publisher, by accepting the article for publication, acknowledges that the
U.S. Government retains a nonexclusive, paid-up, irrevocable, worldwide license to publish or reproduce the published form of this work, or allow others to do so, for U.S. Government purposes.



The authors thank Daniel Zalkind, Garrett Barter, Pietro Bortolotti, Rafael Mudafort, Andy Platt, and Alan Wright from NREL and Greg Hayman from Hayman Consulting LLC for their contributions to discussing the linearization approaches.

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
