# Peer review of "Influence of Wind Turbine Design Parameters on Linearized Physics-Based Models in OpenFAST"

_Wind Energy Science, 2021_

## Author Response (AR1)

**Response to Reviewer Comments**

Jason M. Jonkman[1], Emmanuel S. P. Branlard[1], John P. Jasa[1]

[1]National Renewable Energy Laboratory, 15013 Denver West Parkway, Golden, CO 80401, USA

*Correspondence to*: Jason M. Jonkman (jason.jonkman@nrel.gov)

5    Thank you for supplying us with a thorough review of WES-2021-73. The comments were valuable and we tried to address them all appropriately. In addition to addressing the comments, the document has been reviewed by a professional editor to improve grammar, etc., which resulted in minor editorial changes throughout the paper.

Here are our responses to the specific comments, with the referee comment in green, our response in black, and changes made to the paper indented black.

10   **Referee #1**

Jonkman et al. describe the development of a methodology to compute the sensitivities in linearized dynamic systems with respect to system parameters. The system parameters are treated separately from the state variables during the linearization, resulting in second-order derivatives. The approach is demonstrated on a SDOF mass-spring-damper in order to make clear the steps and fundamental results. It is then used in a case study, looking at how properties such as the tower materials affect
15   the eigenmodes of a floating offshore wind turbine.

The linear analysis of complex systems like floating wind turbines is essential for a human understanding of the dynamics. Linearization is a straightforward mathematical operation, however there are certainly pitfalls and challenges from a practical standpoint, including computational effort. The development of the procedure for this "two-phase" linearization with respect to state variables and system parameters, including constraints, and the implementation of the equations in an open-source tool
20   for the analysis of fixed and floating wind turbines, represents a significant advance in the state-of-the-art.

The manuscript is well-written, with a clear organization in the development of ideas. I have no objections to the technical content. However, I do have one sigificant objection regarding the writing:

(A) As a paper being published in a scientific journal, the references to previous work in the field are inadequate. In fact, only one reference is given to a paper that is not the work of the authors. The content in the manuscript needs to be placed in its
25   proper context relative to the vast body of existing knowledge. Who else has developed tools for linearization of wind turbines? Well, Bladed and HAWCStab2 have such capabilities, for instance. How do they do the linearization? Do they support some sort of system parameter linearization? If not, then this novelty of your approach should be stressed. Is there a precedent in other related fields (aerospace, mechanics, civil, etc) for treating the linearization of state variables and system parameters in this way?

30   *Author response*: Thank you for the kind words regarding our paper. While this exact topic is not found in literature to the author's knowledge, the topic is related to three fields of work and a short connection to each has been added to the end of the Introduction, with a few key references:

Although this exact topic is not found in literature to the authors' knowledge, it is related to three fields of work. The first is linear parameter-varying (LPV) control, whereby a linear state-space system is described by known functions of parameters and gain scheduling is used to switch between the linear systems within the controller based on the current state of those parameters. In the wind turbine field, this technique has been applied to express a linear state-space model in terms of parameters that drive nonlinearities in the system behaviour (e.g., an effective wind speed or blade-pitch angle [Sundarrajan et al., 2021]). Sundarrajan et al. 2021 took this one step further and parameterized a linear floating wind turbine system used for control design in terms of floating platform mass for a CCD application. This article differs from that application in that the approach we develop could apply to any design parameter of the wind turbine like mass, stiffness, geometry, or aerodynamic and hydrodynamic coefficients.

The second field is the use of the Hessian in optimization applications. The Hessian describes the local curvature of the function in terms of second-order partial derivatives of the function. Knowledge of this curvature can greatly improve the convergence of gradient-based optimization methods (Martins and Ning, 2021). This article differs from that application in that Hessians are used in our theoretical approach to describe how the linear system varies with design parameter variation.

Finally, linearized representations of the underlying nonlinear system are commonly generated from wind turbine physics-based engineering tools other than OpenFAST. For example, linearization capability exists in Bladed (DNV GL, 2018) and HAWCStab2 (Hansen et al., 2017), which are other popular physics-based engineering tools used in wind turbine design. However, to the authors' knowledge, the linear models generated from these tools only apply to a fixed set of design parameters, highlighting the novelty of the methods explored in this article.

DNV GL: *Bladed User Manual Version 4.9*. URL: https://www.dnv.com/services/wind-turbine-design-software-bladed-3775. Bristol UK: Garrad Hassan & Partners Ltd, 2018.

Hansen, M. H.; Henriksen, L. C.; Tibaldi, C.; Bergami, L.; Verelst, D.; and Pirrung, G.: *HAWCStab2 User Manual*. URL: https://www.hawcstab2.vindenergi.dtu.dk/. Roskilde, DK: Technical University of Denmark (DTU) Wind Energy, 2017.

Martins, J. R. R. A. and Ning, A.: *Engineering Design Optimization*. URL: http://websites.umich.edu/~mdolaboratory/pdf/Martins2021.pdf. Joaquim R. R. A. Martins and Andrew Ning, 2021.

Sundarrajan, A. K.; Hoon Lee, Y.; Allison, J. T.; and Herber, D. R.: "Open-Loop Control Co-Design of Floating Offshore Wind Turbines Using Linear Parameter-Varying Models," *American Society of Mechanical Engineers (ASME) International Design Engineering Technical Conferences and Computers and Information in Engineering Conference (IDETC/CIE2021), 17–20 August 2021, Virtual, Online* [online proceedings]. URL: https://www.engr.colostate.edu/~drherber/files/Sundarrajan2021c.pdf, DETC2021-67573, 2021.

Minor comments:

(B) The reference to Herber and Allison's paper contains errors, both in the title (missing "Solution"; "Problems" instead of "Problem") and publication year (should be 2019).

*Author's response*: Thanks for noticing these typos; this reference is now fixed:

Herber, D. and Allison, J.: "Nested and Simultaneous Solution Strategies for General Combined Plant and Control Design Problems," *ASME Journal of Mechanical Design*, Vol. 141, No. 1, January 2019, https://doi.org/10.1115/1.4040705, 2019.

**Referee #2**

This paper describes a method to study the effects of variations in design parameters on a linearized system and an operating point (or sets of linearized systems and operating points). The paper provides an example using a mass-spring-damper system to illustrate this method and also provides a case study using direct evaluation and interpolation techniques for parameter variations. We recommend some major and minor revisions to improve the quality of this article.

General/Major Comments:

An example including (equality) constrained algebraic states and an explanation if the methodology can be extended to inequality constraints would be helpful. General references or examples within the context of the wind turbine system would be useful throughout the paper. As an example, stating and elaborating (possibly with references) that "Mooring line tensions are representative of an algebraic constraint state because of <explain>" would be beneficial to the reader and provide context for the various aspects of the turbine that are being linearized.

*Author's response*: The semi-explicit DAE of index 1 that is being considered in this paper does not support inequality constraints. The mass-spring-damper example is meant to be a very simple and intuitive example, but it does have interesting observations (section 3.4) that are not immediately obvious. We feel that adding complexities like an algebraic constraint would complicate the example, rendering the results less illustrative. That said, we have now added examples of continuous states, constraint states, inputs, outputs, and parameters in wind turbine dynamics in the paragraph following Eq. (1):

> …are shown evaluated at time, $t$. In wind turbine dynamics, continuous states may include displacement and velocity of the structure, constraint states may include quasi-steady induction in blade-element/momentum theory, inputs and outputs may include motions and loads, and parameters may include mass, stiffness, geometry, and aerodynamic and hydrodynamic coefficients. A more exhaustive list of example continuous states, constraint states, inputs, outputs, and parameters in wind turbine dynamics are given in Table 2 of Jonkman, 2013.

The use of large matrices throughout the text is tedious. It would not detract from the impact of the paper to simplify them into a more concise format and/or move many of the full matrices to an Appendix. Further, it might also be better to move the entirety of Section 2.3 to an Appendix as the examples used in the rest of the paper do not depend on the results of this section.

*Author's response*: To simplify the equations presented in Sect. 2.2 and 2.3, and to avoid using tensor notation, a "loose notation" has now been introduced, whereby premultiplication of a Hessian by a row vector is a matrix. An example is now given in a new Eq. (8) just after Eq. (7) and used in subsequent equations, which resulted in the renumbering of all equations after Eq. (7):

> To avoid using tensor notation, a "loose notation" is introduced, whereby premultiplication of a Hessian by a row vector is a matrix. As an example, the premultiplication by the transpose of the parameter perturbation vector of the Hessian of the continuous-state functions with respect to parameters and continuous states evaluated at an operating point is outlined in Eq. (8).

$$\Delta p^T \left.\frac{\partial^2 X}{\partial p \partial x}\right|_{op} = \begin{bmatrix} \underset{1\times N_p}{\Delta p^T} \left.\underset{N_p \times N_x}{\frac{\partial^2 X_1}{\partial p \partial x}}\right|_{op} \\ \underset{1\times N_p}{\Delta p^T} \left.\underset{N_p \times N_x}{\frac{\partial^2 X_2}{\partial p \partial x}}\right|_{op} \\ \vdots \\ \underset{1\times N_p}{\Delta p^T} \left.\underset{N_p \times N_x}{\frac{\partial^2 X_{N_x}}{\partial p \partial x}}\right|_{op} \end{bmatrix}_{N_x \times N_x} \qquad (8)$$

Including some sort of nonlinearity in the dynamics of the example problem in Section 3 would be useful; perhaps a nonlinear dashpot. Moreover, having a non-zero steady-state goal might also be helpful.

*Author's response*: See related comment about the mass-spring-damper model above. The mass is already nonlinear in the first-order system of equations. The steady-state operating does have terms that are nonzero; although the velocity and acceleration are zero in the steady state condition, as expected.

Better justification/discussion of the assumptions made when simplifying the equations would enable readers to more clearly understand the methodology presented. Again, references to the wind turbine system would be appropriate in these justifications. Some instances where further justification and explanations would be useful include:

In equation (9), $\Delta u|op$ is set to zero, which might not always be the case. Further explanation/justification is needed here. This also conflicts with the fact that inputs of one module can be outputs of another module.

*Author's response*: Good point. A footnote has been added to further explain this assumption, in the context of module interactions:

* This assumption is valid for an isolated module, uncoupled from other modules. For module interactions in coupled OpenFAST solutions, whose theoretical details are outside the scope of the present article, module-level inputs are derived from module-level outputs through algebraic constraints. So, the change in module-level input operating point can be derived from the change in module-level output operating point similar to how algebraic constraints are eliminated in Sect. 2.3.

Equation (10) is derived from Equation (8) without giving insights why this path is taken and what is achieved in doing so.

*Author's response*: Equation (10) (now (11)) describes how the state and output operating points vary with parameter variations. This has now been stated explicitly in the paragraph following Eq. (11):

Equation (11) describes how the state and output operating points vary with parameter variations. The final expressions…

It is not immediately clear whether the assumption that |dZ/dz| is nonzero is a strong assumption or trivial, especially considering that no example is given of Z in the illustrative example.

*Author's response*: The requirement that the determinant of this Jacobian be nonzero is summarized in the paragraph preceding Eq. (4). More details are explained in reference Jonkman, 2013, which has now been cross referenced in this paragraph:

…and is bounded in the neighbourhood around a solution. These details are discussed more in Jonkman, 2013.

Include more references to the existing literature with respect to linearization tools and methodologies for wind turbine systems.

*Author's response*: See our response to a similar comment made by Referee #1 above.

Additional/Specific Comments:

Make it clear that the example analytical parameter study in Section 3.2 is not addressed in theory and is instead a "baseline" fully exact representation. The definitions of $A(\Delta p)$ etc. labeled "exact" do not match the definitions given in Section 2.3, so either the definition should be made more generally, or they should not be referred to as the same variable.

*Author's response*: Good point. The authors think it is important to show the relation between the Hessian-based approach and the exact parameter dependence. To make this a bit clearly, the "exact" label has been moved from under the equals signs to under the variable names in the equations of Sect. 3.2, which can then be thought of as different, but similar, variables, and a statement has been added before the equations:

…as shown in Eqs. (20) and (21). Note that Eqs. (20) and (21) are the exact analytical expressions of the parameter-dependent linear state matrices and operating points, not derived from the theory of Sect. 2.

The equations presented in the summary observations (equation (24) in Section 3.4) should be presented in the relevant section earlier.

*Author's response*: This equation is tied to the last observation, and so, would be out of place in Sect. 3.3. However, similar to the moving of the "exact" label in the prior comment, the "neglecting…" label has been moved from under the equals signs to under the variable names.

The point in the Observations section (Section 2.4) that, given the presences of constraints, "much of the algebraic manipulation to define the parameterized linear state-space matrices must be implemented in a post-processing step" is very relevant to Section 2.3. As such, we would recommend that this remark should be made in the introduction to Section 2.3.

*Author's response*: This point has now been added in the paragraph after the equations in this section.

Note that that the constraints cannot be algebraically eliminated until the parameter perturbations are explicitly set; this means that while the Jacobians and Hessians can be computed based only on knowledge of the parameter operating point, much of the algebraic manipulation to define the parameterized linear state-space matrices must be implemented in a postprocessing step (once the parameter perturbations are explicitly set).

160    It is not clear, particularly to readers not familiar with OpenFAST, what the line "The first three items in this summary deterred us from implementing the theoretical approach outlined here directly within OpenFAST" implies. It reads as though the testing that the authors claim to have carried out in the abstract was not in fact executed, even though wind turbine simulations were indeed completed via WEIS.

*Author's response*: A similar statement about not implementing the theoretical approach directly within OpenFAST was made
165    in the Introduction. We felt it was important to outline the theoretical development regardless because of the observations that can be intuited by the approach. But to reiterate this point, we added a statement after the quoted paragraph:

The first three items in this summary deterred us from implementing the theoretical approach outlined here directly within OpenFAST. As such, alternative methods have been implemented in WEIS, which calls OpenFAST, instead; see Sect. 4.

170    Some numbered equations with multiple lines are difficult to distinguish from adjacent numbered equations. Either combine some lines in these equations so that it is clearer which equation numbers correspond to those equations (and also so that they take up less vertical space), or separate equation groups with text, or enumerate each equation line (e.g., (19a), (19b), etc.).

*Author's response*: Equation enumerations ((a), (b), etc.) were previously added where important to clarify separations. To avoid potential mix-ups between different equation numbers, white space has now been added between Eqs. (14a), (14b), and
175    (15), between Eqs. (12) and (13), between Eqs. (18) and (19), between Eqs. (20) and (21), and between Eqs. (23) and (24).

Corrections:

Equation (4):

typo – should be u instead of y

*Author's response*: Thanks for noticing this typo.  The y has been changed to u.

180    the equation is referenced several lines before it appears in the text which is confusing

*Author's response*: There are a few places where text-based references to equation appear several lines before the equation, but we felt it was clearer to maintain continuity in a text-based paragraph rather than breaking up the text into separate paragraphs.

the note below the equation contains grammatical errors

185 *Author's response*: We are not sure exactly what grammatical error the referee was referring to. That said, a full editorial review of the paper by a professional editor has been completed, resulting in minor editorial changes throughout the paper.